# Enhanced Interfacial Adhesion of Polylactide/Poly(ε-caprolactone)/Walnut Shell Flour Composites by Reactive Extrusion with Maleinized Linseed Oil

**DOI:** 10.3390/polym11050758

**Published:** 2019-04-30

**Authors:** Sergi Montava-Jordà, Luis Quiles-Carrillo, Nuria Richart, Sergio Torres-Giner, Nestor Montanes

**Affiliations:** 1Technological Institute of Materials (ITM), Universitat Politècnica de València (UPV), Plaza Ferrándiz y Carbonell 1, 03801 Alcoy, Spain; sermonjo@mcm.upv.es (S.M.-J.); nuriaricharts@gmail.com (N.R.); nesmonmu@upvnet.upv.es (N.M.); 2Novel Materials and Nanotechnology Group, Institute of Agrochemistry and Food Technology (IATA), Spanish National Research Council (CSIC), Calle Catedrático Agustín Escardino Benlloch 7, 46980 Paterna, Spain

**Keywords:** PLA, PCL, green composites, multi-functionalized vegetable oils, reactive extrusion, waste valorization

## Abstract

Novel green composites were prepared by melt compounding a binary blend of polylactide (PLA) and poly(ε-caprolactone) (PCL) at 4/1 (wt/wt) with particles of walnut shell flour (WSF) in the 10–40 wt % range, which were obtained as a waste from the agro-food industry. Maleinized linseed oil (MLO) was added at 5 parts per hundred resin (phr) of composite to counteract the intrinsically low compatibility between the biopolymer blend matrix and the lignocellulosic fillers. Although the incorporation of WSF tended to reduce the mechanical strength and thermal stability of PLA/PCL, the MLO-containing composites filled with up to 20 wt % WSF showed superior ductility and a more balanced thermomechanical response. The morphological analysis revealed that the performance improvement attained was related to a plasticization phenomenon of the biopolymer blend and, more interestingly, to an enhancement of the interfacial adhesion of the green composites achieved by extrusion with the multi-functionalized vegetable oil.

## 1. Introduction

Replacement of fossil feedstocks with renewable ones at an equivalent cost is one of the main endeavors of modern plastics industry [1]. One possible strategy to reduce current cost of bioplastics deals with the incorporation of agro-food residues due to their huge availability and low price. Moreover, they also represent a highly sustainable option for waste valorization. The combination of biopolymers and natural fillers (e.g., lignocellulosic fillers) results in the development of the so-called “green composites” [2]. This term indicates that the composite as a whole, that is, both matrix and reinforcement, originates from renewable resources. Resultant green composites offer several environmental advantages over traditional polymer composites such as reduced dependence on non-renewable energy/material sources, lower greenhouse gas and pollutant emissions, improved energy recovery, and end-of-life biodegradability of components [3] as well as potential reduction of both product density and energy requirements for processing [4]. As a result, several lignocellulosic fillers derived from food, agricultural, marine, and industrial wastes have been successfully incorporated into different biopolymer matrices such as almond husk [5,6,7], coconut fibers [8], orange peel [9], rice husk [10,11], recycled cotton [12], peanut shell [13], and posidonia oceanica seaweed [14]. Moreover, the incorporation of cellulosic fillers can also greatly favor disintegration in compost soils [15,16].

In this context, the use of polylactide (PLA) as the matrix in green composites is currently gaining a great importance [17]. PLA is obtained from lactide derived from starch fermentation and it is also fully biodegradable when introduced into an industrial composting plant. Indeed, PLA is nowadays considered the front runner in today’s bioplastics market with an annual consumption of nearly 140,000 tons [18] and a market projection of 800,000 tons by 2020 [19]. Its mechanical properties are comparable to those of polystyrene (PS) and it is increasingly used in rigid packaging applications [20]. Usually the application of PLA is limited by its poor ductility, low heat deflection temperature (HDT), and relatively high cost [21]. In particular, PLA rapid physical aging results in a brittle material with high stiffness and low-impact resistance [22,23]. Blending with elastomeric biopolymers is a well-known approach to overcome the high fragility of PLA [24]. The blending process can be carried out by using conventional machinery, which is an important aspect for industry since no expensive investment is necessary. In this context, poly(ε-caprolactone) (PCL) is a semi-crystalline and linear aliphatic polyester with low glass transition temperature (T_g_) (around −60 °C) that result in highly flexible articles [25]. Although PCL is of petrochemical origin, it is biodegradable in nature due to the susceptibility of its aliphatic ester linkage to hydrolysis [26]. However, PCL shows a relatively low melting temperature (T_m_), in the 59–64 °C range, which restricts its use as mono-component in most packaging applications. Thus, the use of PLA/PCL blends have stimulated extensive research on their potential in rigid packaging due to the improvement attained in impact strength [27]. Nevertheless, detailed studies conducted on these biopolymers have proved that they are immiscible and compatibilization is habitually needed to achieve the properties required for specific applications [28].

Walnut (*Juglans regia* L.) is an important crop that is cultivated throughout the world’s temperate regions for its edible nuts. Worldwide walnut production reached approximately 7.25 million tons in 2016, being China, USA, Iran, and Turkey the main producers [29]. Whereas the walnut kernel is highly interesting for the food industry, the shell has no economic value or industrial usage. Walnut shell comprises approximately 30% of the total walnut production [30], thus representing an annual agricultural waste that accounts for around 2.17 million tons. It is mainly composed by two-thirds holocellulose (hemicellulose and cellulose) and one-third lignin [31]. Since burning causes serious environmental issues [32], a few scholars have recently used walnut shell to produce polymer composites [33,34,35,36,37,38]. However, the incorporation of walnut shell into thermoplastics yields the formation of particle aggregates and results in a deleterious effect on the ductile properties of the composites [39]. This impairment is mainly derived from the poor interfacial adhesion of lignocellulosic fillers with most polymer and biopolymer matrices due to their low chemical affinity [40]. In this context, grafting by means of coupling agents or compatibilizers enhances biopolymer matrix–lignocellulosic filler interactions that then contributes to mitigating fragility decrease [41].

Maleinized linseed oil (MLO) is a cost-competitive cross-linker that is industrially prepared from linseed oil, a natural product that is extracted from the oilseed flax (*Linum usitatissimum* L.) plant. MLO shows one of the highest unsaturation levels amongst common vegetable oils, comparable only to tung oil, thus leading to a highly versatile additive, ripe for chemical functionalization [42]. Therefore, the maleinization process provides multiple maleic anhydride (MAH) functionalities to the MLO structure, which could easily react thereafter with molecules containing hydroxyl (–OH) groups [43,44]. Additionally, small amounts of MLO can also act as a sustainable plasticizer for PLA-based materials, allowing chain motion and improving processing conditions, thermal stability, and ductility [45]. As a result, MLO and other multi-functionalized vegetable oils can be used to enhance compatibility by reactive extrusion (REX) in immiscible or low miscible biopolymer blends and green composites based on biopolyesters and fillers with polar groups. In addition, due to their natural origin, MLO represents an environmentally friendly solution to improve industrial formulations and it can positively contribute to the development of sustainable polymer technologies.

This paper aims to continue and extend of our earlier research work dealing with MLO as a reactive additive to enhance compatibility in immiscible or low miscible green composites based on PLA [6]. The previously followed methodology was applied on PLA/PCL binary blends filled with particles of agro-food waste derived walnut shell to develop green composites with enhanced ductility. The mechanical, morphological, thermal, and thermomechanical properties as well as the water uptake of the resultant green composites were characterized and evaluated in relation to the amount of lignocellulosic filler added and the co-addition of the multi-functionalized vegetable oil.

## 2. Materials and Methods

### 2.1. Materials

Commercial PLA Ingeo™ biopolymer 6201D was provided by NatureWorks (Minnetonka, MN, USA). This PLA resin has a density of 1.24 g/cm^3^ and a melt flow rate (MFR) of 15–30 g/10 min (210 °C, 2.16 kg), which makes it suitable for injection molding. Regarding PCL, a Capa^TM^ 6800 commercial grade was supplied by Perstorp UK Ltd. (Warrington, UK). This PCL resin presents a density of 1.15 g/cm^3^ and a melt flow index (MFI) of 2–4 g/10 min (160 °C, 2.16 kg).

Walnut shell flour (WSF) was supplied by Bazar al andalus (Granada, Spain). According to the manufacturer, the shells were gently separated from the dry fruit and industrially ground with a high-speed rotary cutting mill to achieve a mean particle size lower than 100 µm. Figure 1 shows the walnut shells and the resultant WSF in powder form.

MLO was obtained from Vandeputte (Mouscron, Belgium) as VEOMER LIN. This oil has a viscosity of 1,000 cP at 20 °C and an acid value of 105–130 mg potassium hydroxide (KOH)/g.

### 2.2. Reactive Extrusion

Prior to manufacturing, the biopolymer pellets were dried to minimize their water content in a MDEO dehumidifier from Industrial Marsé (Barcelona, Spain). Drying was performed at 60 °C and 45 °C for PLA and PCL, respectively, both for 36 h. The WSF particles were dried at 100 °C for 48 h.

REX was carried out in a co-rotating twin-screw extruder from Construcciones Mecánicas Dupra, S.L. (Alicante, Spain). The screws presented a diameter of 25 mm and length (L) to diameter (D) ratio, that is, L/D, of 24. All materials were fed through the main hopper, being previously pre-homogenized in a zipper bag. The PLA/PCL ratio was fixed at 4/1 (wt/wt) according to previous findings [27], whereas the amount of WSF in the binary blend varied in the 10–40 wt % range. The MLO content was set at 5 parts per hundred resin (phr) of composite based on our recent research performed using this multi-functionalized vegetable oil [6]. A neat PLA sample, an unfilled PLA/PCL binary blend sample, and a green composite sample at 40 wt % WSF without MLO were also produced as control materials. The temperature profile, from the hopper to the die, was set as follows: 170–175–180–185 °C. Residence time was approximately 1 min, achieved for an output of ~5 kg/h and a speed of the screws during extrusion of 20 rpm. The materials were extruded through a round die to produce strands that were, thereafter, pelletized using an air-knife unit. Table 1 summarizes and codifies the set of formulations prepared.

The compounded pellets were shaped into pieces by injection molding in a Meteor 270/75 from Mateu and Solé (Barcelona, Spain). The temperature profile was 165 °C (hopper), 170 °C, 175 °C, and 180 °C (injection nozzle). A clamping force of 75 tons was applied while the cavity filling and cooling time were set at 1 s and 10 s, respectively. Pieces with a thickness of ~4 mm were produced.

### 2.3. Material Characterization

#### 2.3.1. Mechanical Tests

Tensile tests were carried out in a universal test machine Elib 50 from S.A.E. Ibertest (Madrid, Spain) following the guidelines of ISO 527-1:2012. The selected load cell was 5 kN while the cross-head speed was 2 mm/min. Shore D hardness values were measured with a 676-D durometer from J. Bot (Barcelona, Spain) following ISO 868:2003. Toughness was evaluated by the standard Charpy’s test with a 6-J pendulum from Metrotec S.A. (San Sebastián, Spain) as suggested by ISO 179-1:2010. All specimens were tested at room conditions, that is, 23 °C and 50% relative humidity (RH). At least six samples for each material were tested.

#### 2.3.2. Morphology

The fracture surfaces after the impact tests were observed by field emission scanning electron microscopy (FESEM) using a ZEISS ULTRA 55 model from Oxford Instruments (Abingdon, UK). The samples were previously coated with an ultrathin metallic layer (Au-Pd alloy) to provide electrical conductivity. This process was conducted in vacuum conditions inside a sputter chamber EMITECH mod. SC7620 provided by Quorum Technologies (East Sussex, UK). 

#### 2.3.3. Thermal Analysis

The main transition temperatures and enthalpies were obtained by differential scanning calorimetry (DSC) in a Mettler-Toledo 821 calorimeter (Mettler-Toledo, Schwerzenbach, Switzerland). The sample size ranged from 5 mg to 7 mg and it was placed in standard aluminum crucibles (40 µL). A temperature program based on three stages was performed: first heating from 30 °C to 180 °C, cooling from 180 °C to −50 °C, and second heating from −50 °C to 300 °C. The heating/cooling rates were set at 10 °C/min for all three stages. The main thermal parameters were obtained from the second heating program. All the DSC runs were carried out in inert atmosphere using a flow-rate of nitrogen (N_2_) of 66 mL/min. Measurements were performed in triplicate.

Thermal decomposition was studied by thermogravimetric analysis (TGA) in a TGA/SDTA 851 thermobalance from Mettler-Toledo using a weight sample of 5–8 mg and standard alumina crucibles (70 µL). The thermal program was set from 30 °C to 700 °C at a constant heating rate of 20 °C/min in air. Samples were tested in triplicate.

#### 2.3.4. Thermomechanical Tests

Dimensional stability was carried out by dynamic mechanical thermal analysis (DMTA) in a DMA1 from Mettler-Toledo in the temperature range between −90 °C and 80 °C on injection-molded samples sizing 10 × 7 × 1 mm^3^. The test was carried out in single cantilever bending conditions at a frequency of 1 Hz, with a heating rate of 2 °C/min, and with a deformation control of 10 µm. Samples were evaluated in triplicate.

#### 2.3.5. Water Uptake Measurements

The evolution of the water absorption was followed for a period of 12 weeks according to ISO 62:2008. Injection-molded samples of 4 × 10 × 80 mm^3^ were immersed in distilled water at 23 ± 1 °C. The samples were taken out and weighed weekly, after removing the residual water with a dry cloth, using an AG245 analytical balance from Mettler-Toledo with a precision of ± 0.1 mg. Measurements were performed in triplicate.

### 2.4. Statistical Analysis

The mechanical, thermal, and thermomechanical properties were evaluated through analysis of variance (ANOVA) using STATGRAPHICS Centurion XVI v 16.1.03 from StatPoint Technologies, Inc. (Warrenton, VA, USA). Fisher’s least significant difference (LSD) was used at the 95% confidence level (*p* < 0.05). Mean values and standard deviations were also reported.

## 3. Results and Discussion

### 3.1. Morphology of WSF Particles

Figure 2 shows the FESEM images of the as-received WSF particles taken at low (Figure 2a) and high magnification (Figure 2b) as well as their size histogram (Figure 2c). The low-magnification FESEM micrograph displays a general view of the walnut shell fillers obtained after the grinding and sieving processes, shown in previous Figure 1. One can observe that the resultant ground particles of walnut shell presented a mean size of ~25 µm with a relatively homogeneous size distribution in the 5–50 µm range. Most of the particles were characterized by having a spherical shape though some aggregates as well as flat and long rod-like particles were also observed. Detail of the particle surface of a single walnut shell particle can be seen in the high-magnification FESEM image. The micrograph revealed that the particles were irregular in shape and presented a rough surface, more likely resulting from the crushing process due to the high hardness of this type of filler. Some granular features can also be observed, which resemble the original grainy and wavy structure of walnut shell. A similar morphology for WSF particles has been reported elsewhere [33,36].

### 3.2. Mechanical Properties of PLA/PCL/WSF Composites

Table 2 summarizes the mechanical properties of the injection-molded pieces of neat PLA, PLA/PCL blend, and its green composites with WSF containing MLO. In relation to the tensile properties, the neat PLA piece showed characteristic properties of a rigid article with values of Young’s module and maximum strength of 1101.1 MPa and 61.1 MPa, respectively, and an elongation at break of 8.9%. The incorporation of 20 wt % of PCL into PLA induced a reduction of the Young’s module and maximum strength, showing values of 943.3 MPa and 50.8 MPa, respectively, whereas the elongation at break increased up to a value of 10.4%. Similarly, hardness decreased from 81.4 to 76.4 and impact strength increased from 24.3 kJ/m^2^ to 26.1 kJ/m^2^. In this context, it has been reported that flexible PCL domains can be finely dispersed into the rigid PLA matrix, enhancing ductility and toughness without compromising biodisintegration of PLA [46]. Similar results were observed previously by Simoes et al. [47] for PLA/PCL binary blends. However, the improvement observed in the ductile properties was relatively low, which can be ascribed to the immiscibility between the two biopolymers in the blend [21].

The dual incorporation into the PLA/PCL blend of different contents of WSF and 5 phr MLO produced some remarkable changes in the mechanical performance of the pieces. In all cases, the composites presented lower mechanical strength since the resultant pieces showed lower values of tensile modulus and maximum tensile strength. In particular, the tensile modulus was reduced to values in the 700–850 MPa range whereas tensile strength remained in the range of 10–25 MPa. Interestingly, a remarkable increase in the ductile properties was attained for the pieces containing 10–20 wt % WSF with MLO. In particular, the elongation at break increased up to values of 18.7% and 16.1% for the PLA/PCL blends filled with 10 wt % and 20 wt % WSF, respectively, which represents a percentage increase of approximately 80% and 55%. At higher WSF contents, however, the materials showed a notorious decrease in the elongation at break showing values in the 2–3% range. One can also observe that the composite sample filled with 40 wt % WSF without MLO presented the lowest mechanical performance in terms of both strength and ductility. In relation to hardness, the presence of MLO had a very low influence and all composite pieces showed similar values, in the 74–78 range, being slightly higher for the composites filled with the highest WSF contents. At low filler contents, that is, 10 wt % and 20 wt % WSF, the impact-strength values were kept in the 22–25 kJ/m^2^ range, relatively similar to the unfilled PLA/PCL blend. Once more, WSF contents of 30 wt % and 40 wt % induced a remarkable decrease in toughness, showing values in the range of 5–15 kJ/m^2^.

The effect on the mechanical properties of the single addition of MLO to PLA and its blends was first reported by Ferri et al. [45], who achieved materials with highly improved ductility and similar tensile strength. The enhancement observed in flexibility and ductility was related to the MLO activity as a plasticizer for PLA, decreasing the modulus and yield stress and increasing the strain at break. A similar effect on the mechanical properties of PLA articles has been described for other multi-functionalized vegetable oils. For instance, addition of 2.5–7.5 wt % of acrylated epoxidized soybean oil (AESO) successfully enhanced both elongation at break and impact-absorbed energy of PLA parts, while the tensile and flexural strengths were also maintained or slightly improved [48]. The improvement attained was ascribed to a dual effect of plasticization in combination with a chain-extension and/or cross-linking process of the PLA chains by the highly reactive acrylate and epoxy groups present in the structure of the chemically multi-functionalized oil. A similar mechanism of reactive toughening has also been described recently by our research group for maleinized hemp seed oil (MHO) [49]. Briefly, MHO plasticizes the PLA matrix but simultaneously its various MAH groups can also react with the terminal –OH groups of the PLA chains, generating a macromolecule of higher molecular weight (M_W_) based on a linear chain-extended, branched or even cross-linked structure and, thus, with an improved molecular entanglement to resist mechanical deformation. In other previous study, Xiong et al. [50] reported that the use of low contents of tung oil anhydride (TOA) led to an increase in toughness and impact strength of PLA/starch blends. This effect was based on the reaction between the MAH groups on TOA and the –OH groups on starch, which let the oil molecules to accumulate on the surface of starch and increased compatibility of the blends. Similarly, Wu et al. [51] demonstrated that 10 wt % MAH helped promote bonding between biopolymer blends and tapioca starch via maleation, showing a relatively higher elongation at break. 

In relation to the PLA/PCL/WSF composites, our recent research also demonstrated that 1–5 phr of MLO can successfully improve the mechanical performance of PLA composites containing 30 wt % ASF [6]. Similar to the above, the enhancement achieved was particularly related to a dual compatibilizing effect of plasticization in combination with melt grafting. The latter process was specifically ascribed to the formation of new carboxylic ester bonds (–COO–) through the reaction of the multiple MAH functionalities on MLO with the –OH groups present at the end of the PLA chains and on cellulose on the filler surface. Thus, the fact that the ductility improvement was only observed for the PLA/PCL composites at 10 wt % and 20 wt % WSF suggests that MLO can potentially provide the here-described synergistic effect of plasticization and filler grafting onto the biopolymers at low filler contents. Indeed, these composite pieces also showed the highest values of elongation at break with a minimal loss in mechanical strength. Therefore, the incorporation into PLA/PCL of WSF in contents of up to 20 wt % in combination with MLO resulted in materials with balanced mechanical properties. At high WSF contents, however, the effect of MLO was surpassed by the excessive amount of lignocellulosic fillers. This impairment can be ascribed to the presence of non-grafted fillers and particularly filler agglomerates that potentially acted as a defect rather than reinforcement since these could not efficiently absorb the stresses or prevent the propagation of cracks [6,9]. The impairment was, as expected, particularly relevant for the uncompatibilized green composite piece. This observation can be related not only to the absence of plasticization but also to the lack of strong interactions between the dispersed lignocellulosic particles and the surrounding PLA/PCL matrix. In this context, Chieng et al. [52] reported similar findings in PLA-based composites plasticized with epoxidized vegetable oils (EVOs).

### 3.3. Morphology of PLA/PCL/WSF Composites

Figure 3 gathers the FESEM images of the fracture surfaces obtained after the impact tests of the injection-molded pieces of neat PLA, PLA/PCL blend, and its green composites with WSF containing MLO. Figure 3a shows the fracture of the neat PLA piece, where one can see that the biopolyester is a brittle material, that is, it presents a smooth and relatively flat surface with low roughness. The surface fracture of the binary blend piece, shown in Figure 3b, indicates that the incorporation of PCL into PLA changed the morphology from smooth to rough. One can also observe the formation of an “island-and-sea” morphology based on finely dispersed PCL-rich spherical domains of approximately 1 µm that were well dispersed in the PLA matrix, as reported earlier [46]. Although this biphasic morphology confirms the immiscibility or poor miscibility of PLA with PCL, it is also responsible for improving toughness of PLA since the enclosed PCL microdroplets are able to absorb energy during impact, thus acting as a rubber-like phase dispersed into a brittle matrix [53]. Figure 3c–g shows the fracture surface of the green composite pieces of PLA/PCL/WSF. In the all these FESEM images, one can observe that the WSF particles were relatively well distributed in the PLA/PCL matrix. Interestingly, as the lignocellulosic filler content increased, the dispersed PCL droplets considerably reduced their size or even vanished in the PLA matrix. One can also notice that the co-addition of MLO highly influenced the particle-to-matrix adhesion and also the type of fracture. The improved particle–polymer interaction can be observed in Figure 3c–f by the absence of any gap between the dispersed lignocellulosic particles and the surrounding biopolymer matrix. This interaction allows load transfer between the PLA/PCL matrix and the dispersed WSF particles that, in turn, yields improved ductile properties such as the increase in elongation at break observed above during the mechanical analysis. Plastic deformation provided by MLO was also further supported by the presence of some thin filaments along the PLA/PCL matrix. For the composite pieces with the highest WSF contents, the fracture surfaces were similar but less homogeneous. Although the composites still presented a good particle–polymer continuity, one can also observe a considerably increase in the number of round shape irregularities and/or microvoids on the surface fractures. Indeed, the formation of holes and voids is an indication that some particles were pulled out during fracture, being this phenomenon probably ascribed to particle agglomeration due to the excessive filler content. In this sense, Dufresne et al. [54] described that poor interfacial adhesion and filler agglomeration in green composites habitually lead to numerous irregularly shaped micro-voids or -flaws in the composite structure, which impairs the stress transfer from the matrix to the filler and then reduces the whole mechanical performance of the composite. Finally, in Figure 3g, which corresponds to the composite sample without MLO, one can clearly see the lack of interaction between the lignocellulosic particles and the surrounding PLA/PCL matrix. Material discontinuity can be easily evidenced by the large gap originated during fracture between the particle and matrix, which is responsible for the low fragility of this piece, as it was described above during the mechanical analysis. From a mechanical point of view, the fillers showing this discontinuity are known to act as stress concentrators, that is, they produce the so-called “notch” effect by which high concentrations of stress are formed at the filler interfaces that favor the initiation and spread of cracks, causing a brittle-type fracture [40]. A similar morphology has been observed, for instance, for PLA composites filled with hazelnut shell flour (HSF), in which high filler amounts also promoted micro-crack formation and subsequent growing due to stress concentration during impact conditions [55]. One can also observe that the fracture surface was smoother and more homogenous than those of the MLO-containing composites.

The remarkable change observed in the fracture surface morphology can be ascribed, as described above, to the multiple actions that MLO can potentially provide to the PLA/PCL/WSF composites. Firstly, low contents of MLO can successfully plasticize the PLA-based matrix [56], then leading to lower stresses at the biopolymer–filler interface. Plasticization is based on the fact that the vegetable oil molecules can readily place among the PLA chains, then acting as lubricant-type agents with enhanced influence on chain mobility [52]. Secondly, MLO could also favor miscibility between the PLA and PCL phases due to its multi-functionality. In this regard, it has been shown that the use of multi-functionalized vegetable oils yields reactive compatibilization in binary blends made of PLA with TPS [45,57] and, more recently, with bio-based high-density polyethylene (bio-HDPE) [58]. The compatibilization reported in some of these previous studies was ascribed to the reaction of the multiple functional groups of the vegetable oil with the functional groups of condensation polymers, that is, –OH and carboxyl (–COOH) terminal groups, during extrusion. Finally, MLO can additionally promote melt grafting of the lignocellulosic WSF particles onto the plasticized PLA/PCL matrix, thus leading to an enhanced interfacial adhesion of the composites. The latter effect has been also ascribed to the presence of a various number of MAH groups on the vegetable oil molecules that can react during extrusion with both the –OH groups of the biopolymer and of the lignocellulic fillers to form new –COO– linkages [6]. Similar to this study, Balart et al. [59] previously reported that epoxidized linseed oil (ELO) can effectively improve particle–matrix interaction of PLA/HSF composites by a double action of the multi-functionalized oil. On the one hand, ELO provides a plasticizing effect on the PLA matrix that leads to an improved chain mobility and, on the other hand, it attains melt grafting by a coupling mechanism based on the reaction of the epoxy groups with the –OH groups present on both the biopolymers and the lignocellulosic filler.

### 3.4. Thermal Properties of PLA/PCL/WSF Composites

Figure 4 shows the DSC curves obtained during second heating of the injection-molded pieces of neat PLA, PLA/PCL blend, and its green composites with WSF containing MLO. Table 3 summarizes the main thermal parameters obtained from these curves. In relation to the neat PLA piece, one can see that the biopolymer presented T_g_ at ~63 °C and then it developed cold crystallization at higher temperatures, showing a cold crystallization temperature (T_CC_) at 113 °C. One can also observe that the sample showed two overlapped melting peaks, having a low-intensity first melting point at ~166 °C and a more intense second one at nearly 172 °C. Double melting in polymers is related to a phenomenon of melt recrystallization during heating [60]. According to this model, the melting and recrystallization are competitive in the heating process so that small and imperfect crystals change successively into more stable crystals through a melt-recrystallization mechanism. Both phenomena of cold crystallization and double melting are an indication of restricted crystallization, which has been previously reported for PLA [61]. Cold crystallization typically occurs during the melting process of rapidly cooled PLA samples whereas the melting of the melt-crystallized sample starts from a low temperature due to the sample includes thinner and imperfect crystals. In relation to the DSC curve of the unfilled blend piece, one can observe a small endothermic peak associated to the melting process of the PCL phase, being located at ~58 °C. This melting process overlapped with the glass transition region of PLA so that it was not possible to separate both thermal processes by conventional DSC [46]. Interestingly, the incorporation of PCL shifted the T_CC_ value of PLA to a lower temperature, that is, 99 °C, whereas it reduced slightly T_m_ to ~170 °C and also suppressed the double-melting peak behavior. This observation points out that PCL enhanced crystallization of PLA. One assumes that this effect can ascribed to plasticization by the dispersed molten phase, that is, PCL, which enhanced PLA chains mobility and then their folding [62].

The addition of WSF reduced slightly both the T_CC_ and T_m_ values of PLA in the blend. This effect was more remarkable for the composite piece without MLO, in which T_CC_ and T_m_ were reduced to values below 95 °C and 165 °C, respectively. The lower T_CC_ values obtained suggests that the WSF particles provided a nucleating effect on PLA during cold crystallization but, as T_m_ was also reduced, it is considered that they also impaired the formation of thicker, more perfect, and stable crystals. The reduction observed in the enthalpies of both cold crystallization (∆H_CC_) and melting (∆H_m_) corroborated the latter effect. In this context, Chun et al. [63] observed that coconut shell powder (CSP), which was chemically modified with maleic acid (MA), nucleated the formation of PLA crystals. In another work, Lee et al. [64] showed that the loading of wood flour (WF) in the 10–40 wt % range decreased T_m_’s PLA by 1.5–1.7 °C. The decrease observed in the melting peak of the biopolyester was related to a phenomenon of physical hindrance by the lignocellulosic fillers, which disrupted the chain-folding process of PLA. Therefore, the presence of the WSF particles nucleated the formation of more crystals, though less perfect due to an effect of molecular restriction.

Figure 5 shows the TGA (Figure 5a) and DTG curves (Figure 5b) of the neat PLA, PLA/PCL blend, and the PLA/PCL composites at different WSF contents compatibilized with MLO. Table 4 shows the summarized data obtained from the TGA and DTG curves. The onset of degradation of PLA, measured at the temperature at which a mass loss of 5% was produced (T_5%_), was ~340 °C, whereas thermal degradation (T_deg_) was ~375 °C. One can also observe that PLA degraded in two stages. The main degradation step occurred in the temperature range of 340–390 °C, showing a mass loss of ~96%. A second mass loss, of much lower intensity (<4%), was observed from approximately 450 °C to 500 °C. The corresponding area of this latter zone on the DTG curve is evidenced by the small peak on the right-hand side. The first major peak was due to thermal degradation of the biopolyester from high-M_W_ macromolecules into smaller chain fragments, while the second minor peak can be related to the thermal degradation of the small-M_W_ biopolymer fragments. In this regard, Signori et al. [65] observed two degradation steps during the thermal degradation of PLA under air flow, indicating that thermo-oxidative degradation pathways occurred. It was also determined that weight loss of the second step for PLA was very low (~2.5%). One can also observe that the addition of PCL reduced both T_5%_ and T_deg_ by approximately 10 °C and 5 °C, respectively. In addition, the second peak of mass loss increased to ~7%. In this context, Vilay et al. [66] similarly reported that thermal degradation of PLA/PCL blends were characterized by having two degradation steps, which was ascribed to the existence of two distinct and immiscible phases derived from the PLA matrix and the dispersed PCL phase.

The incorporation of WSF reduced considerably the thermal stability of PLA/PCL. One can observe that the T_5%_ and T_deg_ values decreased markedly as the filler content increased. It is also worth noting that the rate of weight loss of the biopolymers was considerably reduced, which suggests that the fillers exerted a mass transport barrier to the volatiles produced during their thermal decomposition. One assumes that this impairment is directly related to the relatively low thermal stability of the lignocellulosic filler. In particular, the TGA curve of WSF showed three main weight losses. The first one occurred around 100 °C, showing a mass loss of ~1.9%. It mainly corresponds to the removal of the filler-remained water after drying due to the great tendency of the lignocellulosic materials to absorb moisture [67]. Indeed, it was found that, despite the drying process, the content of water in walnut shells can reach values typically in the range of 1.5–2 wt % [36]. Following the TGA curve of WSF, the main degradation peak started at approximately 195 °C and ended at 345 °C, showing an average mass loss of ~48%. This zone represents the main devolatilization step of biomass pyrolysis and it is referred as the “active pyrolysis zone” since mass loss rate is high [31]. This mass loss can be identified as that of hemicellulose and cellulose decomposition, which are the main components of walnut shell. Both degradation processes involve complex reactions (e.g., dehydration, decarboxylation, among others) as well as breakage of C–H, C–O, and C–C bonds [68]. One can observe that the initiation of the third degradation step overlapped with the end of the second peak and it continued progressively up to temperatures in the 600–700 °C range. Average mass loss of the third peak was determined as ~22% and it is seen as a tailing in both TGA and DTG curves. This zone represents the so-called “passive pyrolysis zone” since the mass loss attained is smaller and the mass loss rate is also much lower compared to that observed in the second zone [31]. This mass loss can be assigned to the thermal decomposition of lignin since it is known to occur slowly in a broad temperature range [69]. 

The impairment observed in the thermal stability is in agreement with previous studies concerning green composites based on nut shells [70,71,72]. The thermal stability reduction reported in these previous studies was mainly ascribed to the low degradation temperature of the lignocellulosic particles. However, both the high shear and frictional forces experienced by the composites during extrusion and, particularly, the remaining water content on the fillers can additionally influence negatively on the thermal stability of the biopolymers. The lowest thermal stability was observed for the PLA/PCL/WSF composite without MLO, in which the T_5%_ and T_deg_ values decreased to approximately 252 °C and 293.5 °C, respectively. Indeed, the same composite prepared with 5 phr MLO showed values of T_5%_ and T_deg_ around 269 °C and 300 °C. Therefore, the MLO addition exerted a positive effect on the overall thermal stability of the green composites. This thermal stability increase in the MLO-containing green composite has been related to the chemical interaction achieved by the multi-functionalized vegetable oil due to the covalent bonds established between the lignocellulosic fillers and the biopolymer matrix [6]. Measured at 700 °C, the sample residual mass grew with the increasing content of the WSF filler, reaching values of up to ~8%.

### 3.5. Thermomechanical Properties of PLA/PCL/WSF Composites

Figure 6 shows the DMTA curves of the injection-molded pieces of neat PLA, PLA/PCL blend, and PLA/PCL/WSF composites using MLO as compatibilizing agent. Table 5 includes the values of storage modulus at −80 °C and 20 °C, as being representative for describing the thermomechanical response of the pieces, and also the estimated T_g_ values from the damping factor (tan *δ*) curves. The evolution of the storage modulus in the temperature range from −90 °C to 80 °C is plotted in Figure 6a. This thermomechanical property is directly related to the stored elastic energy and, consequently, it provides information about stiffness of the pieces. One can observe that the neat PLA piece presented storage modulus values in the 1.50–1.75 GPa range for temperatures below 50 °C. Then, in the 50–70 °C range, the storage modulus showed a decrease of three orders of magnitude, reaching values of ~5 MPa. This change in the thermomechanical response corresponds to the alpha (α)-relaxation of PLA, which is ascribed to its glass–rubber transition and relates to the biopolymer’s T_g_. Slightly higher values of storage modulus were observed in the DMTA curves of the PLA/PCL piece at temperatures below −70 °C. This effect can be ascribed to the fact that, at this temperature, the dispersed PCL phase remains as a vitreous solid and, therefore, it hardens the PLA matrix (~1.8 GPa at −80 °C). However, as PCL started undergoing its glass–rubber transition at around −60 °C, the blend pieces presented a progressive decrease in the storage modulus similar to that observed for neat PLA. This confirms that the presence of PCL increased material’s toughness at room temperature (~1.4 GPa at 20 °C). One can also observe that the incorporation of WSF in combination with MLO further decreased the storage modulus of PLA/PCL, then leading to softer materials. This supports the above-described plasticizing effect provided by MLO during the mechanical analysis and it is also in agreement with our previous study on PLA/AHF composites compatibilized by MLO [5]. Excepcionally, both green composites at 40 wt % WSF showed a remarkable increase in the storage modulus at low temperatures. This result suggests that, at very low temperatures, the thermomechanical performance of the green composite was mainly controlled by the fillers since they were dispersed in high contents in a glassy matrix. Similar observations were reported by Balart et al. [55] in which the addition of lignocellulosic HSF particles, at contents in the 10–40 wt % range, restricted the PLA chain mobility and resulted in stiffer materials. It is also worthy to mention that the highest storage modulus was observed for the MLO-containing green composite at lower temperatures (~2.2 GPa at −80 °C), whereas the non-compatibilized sample showed higher values at higher temperatures (~1.5 GPa at 20 °C). This observation points out that MLO succesfully enhanced filler–matrix adhesion but the plasticizing effect had more influence on the mechanical strenght of the green composites at higher temperatures.

The evolution of tan δ versus temperature is shown Figure 6b. This property relates the ratio of the energy lost, due to viscous behavior, to the energy stored, due to elastic one, in a cyclic deformation. The intense peak located between 50 °C and 80 °C corresponds to α-relaxation of PLA, which is related to its T_g_. In the case of the neat PLA piece, the α-peak was centered at 68.4 °C. This value was similar but slighlty higher than that observed by DSC during the second heating, which may be explained by differences in the test conditions and sample crystallinity. The zoomed inset in the graph shows the α-relaxation of PCL in which one can observe that the unfilled PLA/PCL piece presented a broad and low-intensity peak at −53.4 °C. The incoporation of WSF and MLO interestingly reduced and also shifted both α-relaxation peaks to intermediate temperatures. The peak reduction observed indicates that the presence of both PCL and WSF partially suppressed the relaxation of the PLA chains [73] and this effect intensified as the filler content increased. In relation to the peak displacements, the estimated T_g_ values of the dispersed PCL phase increased to temperatures in the range from approximately −46 °C to −52 °C, whereas those for PLA decreased to the 63–65 °C range. The reduction of the PLA’s T_g_ can be ascribed to the aforementioned process of plasticization by PCL and MLO, which is consistent with the previous mechanical and thermal analyses. This thermomechanical change was, however, also observed for the uncompatibilized green composite. Moreover, the T_g_ values of the PCL phase shifted progressively to higher temperatures with the WSF content. The latter observation suggests that the lignocellulosic fillers also contributed to improving miscibility between both biopolymers. Indeed, partial miscibility can be inferred by the shift of the T_g_ value of one biopolymer toward that of the other biopolymer [74]. This further supports the above-observed SEM micrographs during morphological anaylsys in which the dispersed PCL phase was not discerned in the PLA matrix of the composite pieces filled with high WSF contents. This result, therefore, indicates that the lignocellulosic fillers provided a more optimal processing window during melt mixing to blend PLA and PCL biopolymers. One proposes that the addition of large filler amounts raised thermomechanical stresses during melt processing due to an intensification of filler-to-filler collisions that, in turn, produced lower and more similar viscosities for both biopolymer melts by an effect of melt-shear thinning.

### 3.6. Water Uptake of PLA/PCL/WSF Composites

One of the most important drawbacks of green composites is their tendency to absorb water due to the hydrophilic nature of the incorporated lignocellulosic fillers. The evolution of the water absorption as a function of the immersion time for a period of 12 weeks is shown in Figure 7. The PLA piece reached the equilibrium water uptake at a value of ~0.7 wt %, confirming the intrinsic hydrophobic behavior of PLA. The incorporation of PCL into PLA had a low influence on water absorption since the water uptake of PCL is also very low due to the non-polar nature of this biopolyester. In this sense, Malin et al. [75] showed that PCL, at room temperature, presents a water absorption of nearly 0.6 wt %. The slight increase observed can be then ascribed to the above-described plasticization of PLA by PCL. In the composite pieces, however, the equilibrium water uptake increased significantly, and it was found to depend strongly on the lignocellulosic filler content. In the case of the green composite pieces filled with 10 wt % and 20 wt % WSF, both compatibilized with MLO, water absorption increased with immersion time reaching a plateau after approximately 6 weeks with asymptotic values of approximately 10 wt % and 12.5 wt %. Similar absorption curves but with significantly higher values were observed for the MLO-containing green composites pieces with 30 wt % and 40 wt % WSF, that is, approximately 25 wt % and 34 wt %, respectively. The uncompatibilized green composite piece with 40 wt % WSF absorbed ~23 wt % of water. Therefore, MLO favored water absorption increase, which is in agreement with our previous study of green composites based on PLA [5]. This increase in water absorption is a direct consequence of the plasticizing effect provided by the multi-functionalized vegetable oil since free volume of the PLA matrix is enlarged, favoring water diffusion. It seems therefore that, in terms of water uptake, the use of a different compatibilizer and/or the additional pre-treatment of the filler surfaces to make them more hydrophobic will be necessary to expand the use of the here-prepared green composites in a damp atmosphere.

## 4. Conclusions

The dual incorporation into PLA/PCL of up to 20 wt % WSF and 5 phr MLO successfully yielded composites with improved ductility and a minimal loss in mechanical strength and toughness. The fracture surfaces of the pieces revealed that the MLO co-addition favored the particle-to-matrix adhesion and increased plastic deformation. The incorporation of WSF combined with MLO resulted in a lower size of the dispersed PCL phase into PLA. Moreover, MLO slighlty improved thermal stability and reduced rigidity at room temperature. The improvement achieved was related to an enhancement of the interfacial adhesion of the PLA/PCL/WSF composites based on the multiple actions of MLO during extrusion. Firstly, the vegetable oil molecules induced a plasticization phenomenon of the biopolymer matrix. The lubricating effect of MLO increased biopolymer chain mobility and, thus, yielded lower stresses at the biopolymer–filler interface. Secondly, the presence of MLO in combination to high contents of WSF could also promote certain miscibility for the binary blend of biopolymers that constitute the composite’s matrix. Lastly, the multi-functionality of MLO additionally provided melt grafting of the WSF fillers onto the PLA/PCL matrix. The resultant enhanced particle–biopolymer continuity of the composites allowed better load transfer between the PLA/PCL matrix and the dispersed WSF particles that, in turn, led to materials with higher mechanical and thermal performance. MLO could, however, only provide the multiple actions of plasticization and enhanced interfacial adhesion at moderate WSF contents. At higher contents, the effect of MLO was surpassed by the presence of non-grafted fillers and particularly filler agglomerates that potentially acted as a defect rather than reinforcement.

This study demonstrates that MLO is a multi-functionalized vegetable oil that can be very attractive as a novel additive for the biopolymer industry since it can positively contribute to the performance enhancement of PLA-based composites. The here-developed green composites with improved ductility can easily find applications in rigid packaging (e.g., sustainable trays, bottles, and caps) or in the building and construction industry (e.g., wood boards and docks).

## Figures and Tables

**Figure 1 polymers-11-00758-f001:**
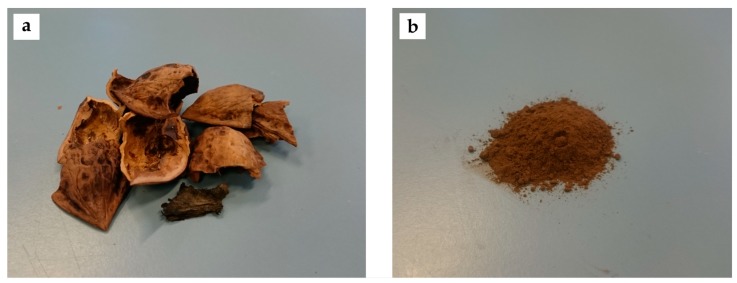
(**a**) Walnut shells; (**b**) walnut shell flour (WSF). Courtesy of Bazar al andalus (Granada, Spain).

**Figure 2 polymers-11-00758-f002:**
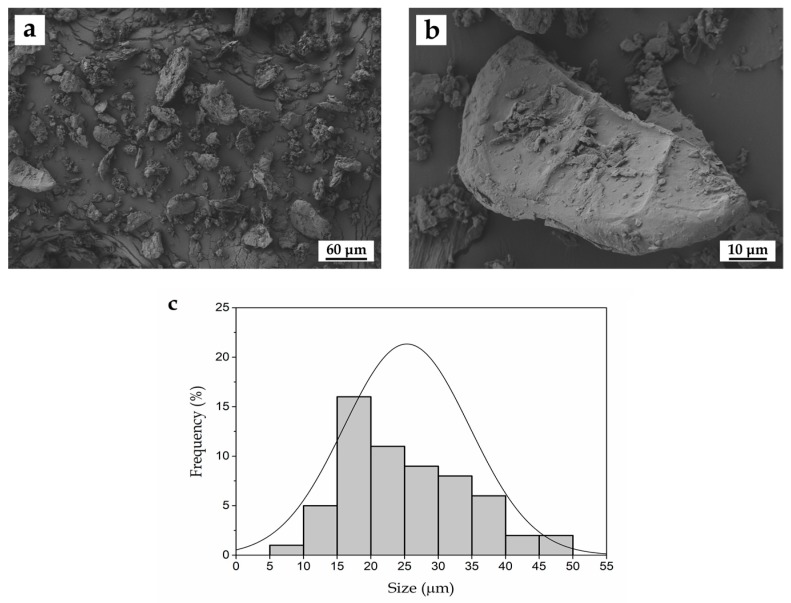
Field emission scanning electron microscopy (FESEM) micrographs of walnut shell flour (WSF) taken at (**a**) 250× and (**b**) 1500× showing scale markers of 60 µm and 10 µm, respectively; (**c**) Particle size histogram of WSF.

**Figure 3 polymers-11-00758-f003:**
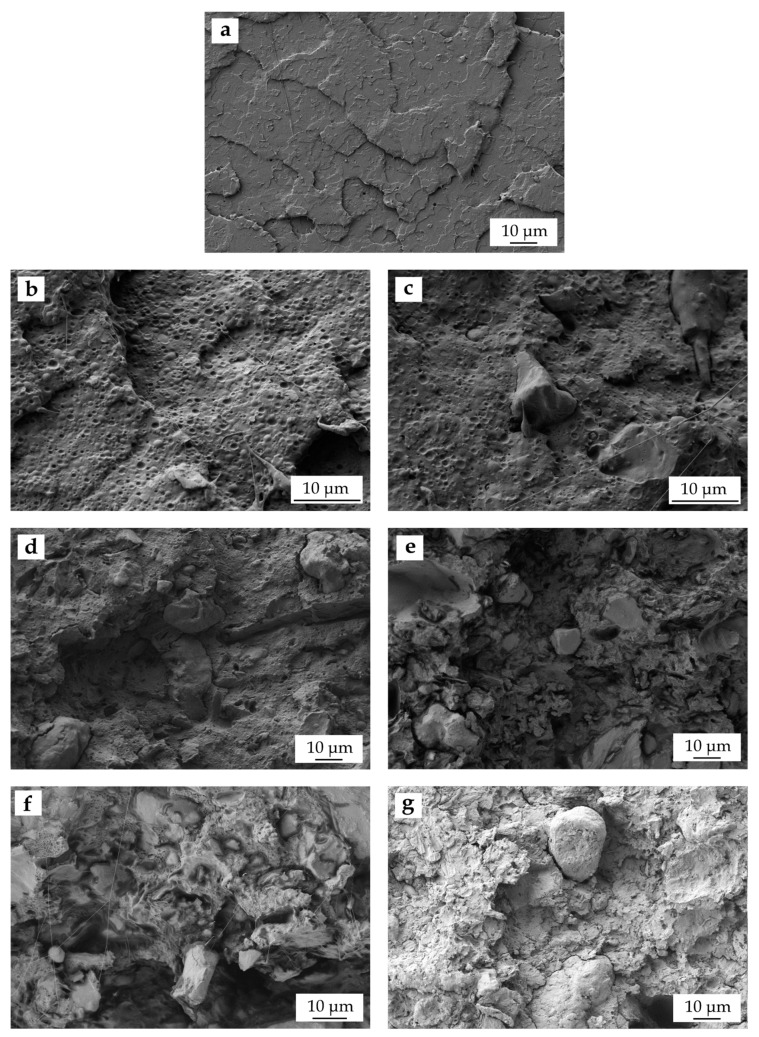
Field emission scanning electron microscopy (FESEM) images of the fracture surfaces of the green composites made of polylactide (PLA), poly(ε-caprolactone) (PCL), walnut shell flour (WSF), and maleinized linseed oil (MLO): (**a**) PLA; (**b**) PLA/PCL; (**c**) PLA/PCL/10WSF + MLO; (**d**) PLA/PCL/20WSF + MLO; (**e**) PLA/PCL/30WSF + MLO; (**f**) PLA/PCL/40WSF + MLO; (**g**) PLA/PCL/40WSF. Images were taken at 1000x and scale markers are of 10 µm.

**Figure 4 polymers-11-00758-f004:**
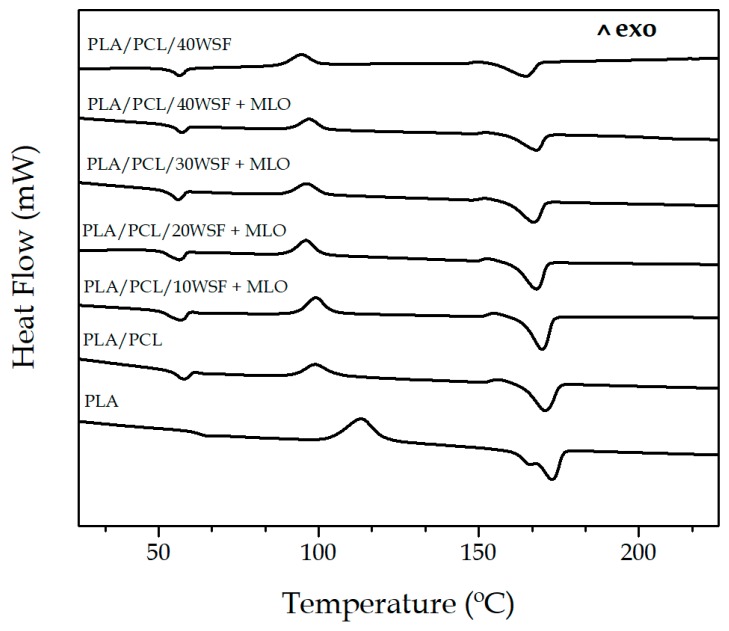
Differential scanning calorimetry (DSC) curves of the green composites made of polylactide (PLA), poly(ε-caprolactone) (PCL), walnut shell flour (WSF), and maleinized linseed oil (MLO).

**Figure 5 polymers-11-00758-f005:**
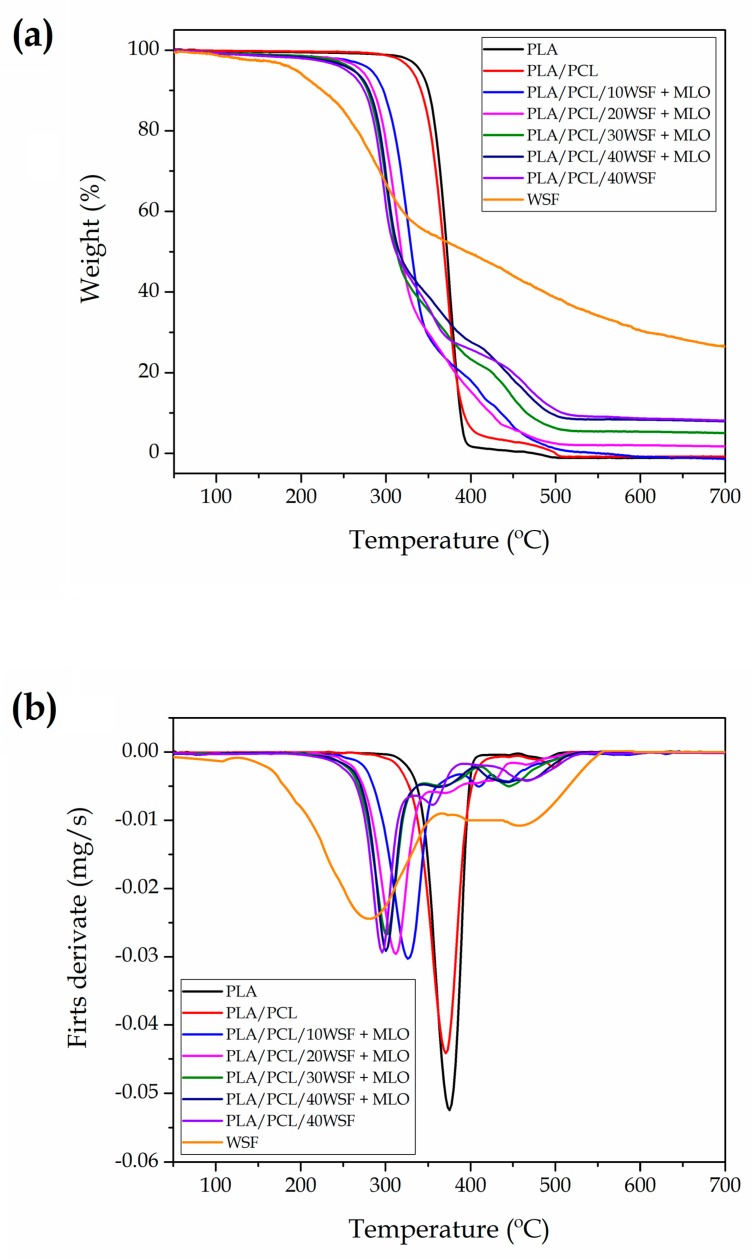
(**a**) Thermogravimetric analysis (TGA) and (**b**) first derivative (DTG) curves of the green composites made of polylactide (PLA), poly(ε-caprolactone) (PCL), walnut shell flour (WSF), and maleinized linseed oil (MLO).

**Figure 6 polymers-11-00758-f006:**
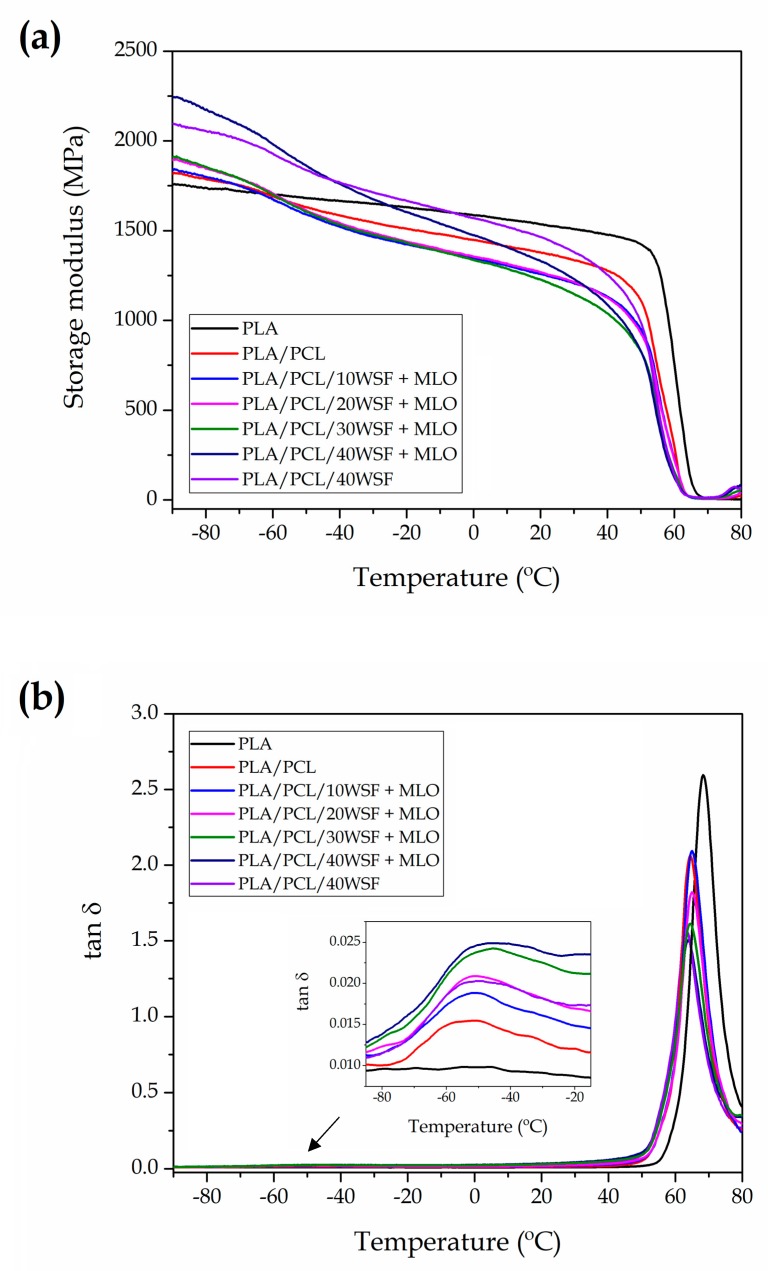
Evolution as a function of temperature of the (**a**) storage modulus (E’) and (**b**) dynamic damping factor (tan *δ*) of the green composites made of polylactide (PLA), poly(ε-caprolactone) (PCL), walnut shell flour (WSF), and maleinized linseed oil (MLO).

**Figure 7 polymers-11-00758-f007:**
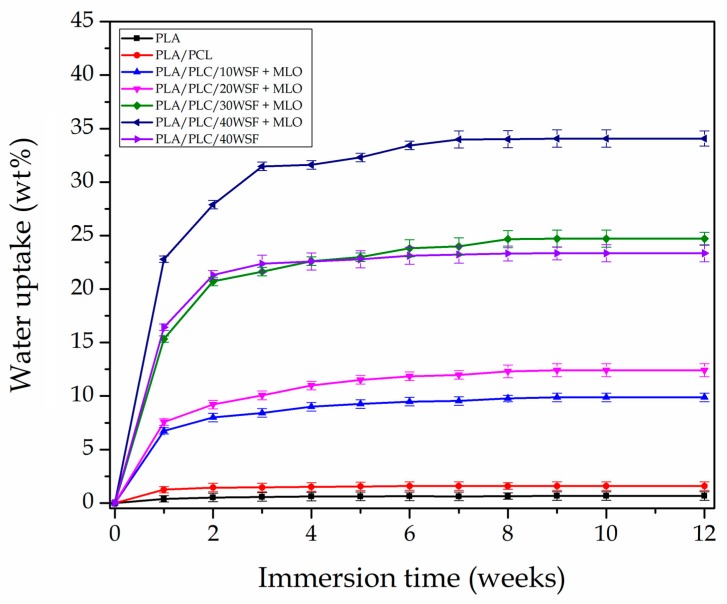
Water uptake of the green composites made of polylactide (PLA), poly(ε-caprolactone) (PCL), walnut shell flour (WSF), and maleinized linseed oil (MLO).

**Table 1 polymers-11-00758-t001:** Summary of compositions according to the weight content (wt %) of polylactide (PLA), poly(ε-caprolactone) (PCL), and walnut shell flour (WSF) in which maleinized linseed oil (MLO) was added as parts per hundred resin (phr) of PLA/PCL/WSF composite.

Sample.	PLA (wt %)	PCL (wt %)	WSF (wt %)	MLO (phr)
PLA	100	0	0	0
PLA/PCL	80	20	0	0
PLA/PCL/10WSF + MLO	72	18	10	5
PLA/PCL/20WSF + MLO	64	16	20	5
PLA/PCL/30WSF + MLO	56	14	30	5
PLA/PCL/40WSF + MLO	48	12	40	5
PLA/PCL/40WSF	48	12	40	0

**Table 2 polymers-11-00758-t002:** Summary of the mechanical properties of the green composites made of polylactide (PLA), poly(ε-caprolactone) (PCL), walnut shell flour (WSF), and maleinized linseed oil (MLO) in terms of: tensile modulus (E), maximum tensile strength (σ_max_), elongation at break (ε_b_), Shore D hardness, and impact strength.

Piece	E (MPa)	σ_max_ (MPa)	ε_b_ (%)	Shore D Hardness	Impact Strength (kJ/m^2^)
PLA	1101.1 ± 50.1 ^a^	61.1 ± 0.8 ^a^	8.9 ± 0.2 ^a^	81.4 ± 1.7 ^a^	24.3 ± 2.1 ^a^
PLA/PCL	943.3 ± 24.2 ^b^	50.8 ± 0.4 ^b^	10.4 ± 1.4 ^a^	76.4 ± 2.1 ^a^	26.1 ± 2.3 ^a^
PLA/PCL/10WSF + MLO	832.9 ± 25.5 ^c^	25.4 ± 0.3 ^c^	18.7 ± 1.7 ^b^	75.0 ± 1.2 ^b^	24.5 ± 2.8 ^a^
PLA/PCL/20WSF + MLO	787.7 ± 35.3 ^d^	21.8 ± 1.1 ^c^	16.1 ± 2.2 ^b^	74.6 ± 1.8 ^b^	22.6 ± 1.9 ^a^
PLA/PCL/30WSF + MLO	723.2 ± 29.6 ^d^	13.4 ± 1.0 ^d^	3.1 ± 0.7 ^c^	75.4 ± 0.5 ^b^	13.1 ± 1.4 ^b^
PLA/PCL/40WSF + MLO	714.5 ± 49.5 ^d^	10.4 ± 1.3 ^d^	2.1 ± 0.5 ^c^	75.9 ± 0.6 ^b^	9.4 ± 1.6 ^c^
PLA/PCL/40WSF	604.8 ± 52.9 ^d^	8.9 ± 2.2 ^d^	1.5 ± 0.1 ^c^	77.8 ± 1.3 ^a^	4.9 ± 0.9 ^d^

^a–d^ Different letters in the same column indicate a significant difference among the samples (*p* < 0.05).

**Table 3 polymers-11-00758-t003:** Main thermal parameters of the green composites made of polylactide (PLA), poly(ε-caprolactone) (PCL), walnut shell flour (WSF), and maleinized linseed oil (MLO) in terms of: normalized enthalpy of cold crystallization (∆H_CC_), cold crystallization temperature (T_CC_), normalized enthalpy of melting (∆H_m_), and melting temperature (T_m_) for the PLA and PCL phases.

Sample	T_m PCL_ (°C)	∆H_m PCL_ (J/g)	T_CC PLA_ (°C)	∆H_CC PLA_ (J/g)	T_m PLA_ (°C)	∆H_m PLA_ (J/g)
PLA	-	-	113.0 ± 0.6 ^a^	31.0 ± 0.8 ^a^	165.9 ± 0.4 ^a^ / 172.2 ± 0.6 ^b^	37.0 ± 0.9 ^a^
PLA/PCL	57.8 ± 0.8 ^a^	8.0 ± 0.3 ^a^	99.1 ± 0.8 ^b^	16.5 ± 0.8 ^b^	170.2 ± 0.7 ^c^	38.5 ± 0.8 ^a^
PLA/PCL/10WSF + MLO	56.5 ± 0.9 ^a^	6.9 ± 0.4 ^b^	98.9 ± 0.7 ^b^	15.8 ± 0.6 ^b^	169.3 ± 0.8 ^c^	33.9 ± 0.6 ^b^
PLA/PCL/20WSF + MLO	56.3 ± 0.7 ^a^	6.7 ± 0.2 ^b^	95.9 ± 0.9 ^c^	15.5 ± 0.6 ^b^	167.5 ± 0.6 ^d^	32.5 ± 0.6 ^b^
PLA/PCL/30WSF + MLO	55.8 ± 1.1 ^a^	5.6 ± 0.3 ^c^	96.0 ± 0.8 ^c^	12.7 ± 0.5 ^c^	166.7 ± 0.7 ^d^	24.9 ± 0.5 ^c^
PLA/PCL/40WSF + MLO	57.0 ± 1.0 ^a^	4.7 ± 0.4 ^d^	96.9 ± 0.7 ^c^	11.3 ± 0.6 ^c^	167.6 ± 0.6 ^d^	20.8 ± 0.6 ^d^
PLA/PCL/40WSF	56.5 ± 0.9 ^a^	4.6 ± 0.5 ^d^	94.6 ± 0.6 ^d^	10.9 ± 0.6 ^d^	164.6 ± 0.5 ^a^	20.5 ± 0.6 ^d^

^a–d^ Different letters in the same column indicate a significant difference among the samples (*p* < 0.05).

**Table 4 polymers-11-00758-t004:** Main thermal parameters of the green composites made of polylactide (PLA), poly(ε-caprolactone) (PCL), walnut shell flour (WSF), and maleinized linseed oil (MLO) in terms of: onset temperature of degradation (T_5%_), degradation temperature (T_deg_), and residual mass at 700 °C.

Sample	T_5%_ (°C)	T_deg_ (°C)	Residual Mass (%)
PLA	340.3 ± 1.2 ^a^	375.3 ± 1.2 ^a^	0.6 ± 0.2 ^a^
PLA/PCL	329.6 ± 1.3 ^b^	369.7 ± 0.9 ^b^	0.3 ± 0.2 ^a^
PLA/PCL/10WSF + MLO	282.4 ± 1.6 ^c^	326.5 ± 1.4 ^c^	1.8 ± 0.7 ^b^
PLA/PCL/20WSF + MLO	270.1 ± 1.1 ^d^	311.6 ± 1.1 ^d^	2.3 ± 0.6 ^b^
PLA/PCL/30WSF + MLO	262.7 ± 0.9 ^e^	302.7 ± 0.6 ^e^	5.5 ± 0.9 ^c^
PLA/PCL/40WSF + MLO	259.8 ± 1.4 ^e^	300.2 ± 0.9 ^e^	7.8 ± 0.7 ^c^
PLA/PCL/40WSF	252.3 ± 1.5 ^f^	293.5 ± 1.1 ^f^	8.2 ± 0.8 ^c^
WSF	194.1 ± 0.5 ^g^	278.5 ± 1.8 ^g^	27.5 ± 1.2 ^d^

^a–g^ Different letters in the same column indicate a significant difference among the samples (*p* < 0.05).

**Table 5 polymers-11-00758-t005:** Main thermomechanical parameters of the green composites made of polylactide (PLA), poly(ε-caprolactone) (PCL), walnut shell flour (WSF), and maleinized linseed oil (MLO) in terms of: storage modulus (E’) measured at −80 °C and 20 °C and glass transition temperature (T_g_) for the PLA and PCL phases.

Sample	E (MPa) at −80 °C	E (MPa) at 20 °C	T_g _PCL__ (°C)	T_g _PLA__ (°C)
PLA	1745 ± 16 ^a^	1536 ± 12 ^a^	-	68.4 ± 0.9 ^a^
PLA/PCL	1782 ± 12 ^a^	1375 ± 11 ^b^	−53.4 ± 1.2 ^a^	64.4 ± 0.7 ^b^
PLA/PCL/10WSF + MLO	1792 ± 13 ^a^	1260 ± 8 ^c^	−51.9 ± 0.9 ^b^	64.9 ± 1.0 ^b^
PLA/PCL/20WSF + MLO	1846 ± 17 ^b^	1275 ± 10 ^c^	−50.6 ± 0.8 ^b^	65.0 ± 0.8 ^b^
PLA/PCL/30WSF + MLO	1850 ± 10 ^b^	1120± 12 ^d^	−46.1 ± 1.1 ^c^	64.4 ± 0.9 ^b^
PLA/PCL/40WSF + MLO	2175 ± 24 ^c^	1330 ± 14 ^e^	−45.8 ± 0.7 ^c^	63.8 ± 0.7 ^b^
PLA/PCL/40WSF	2055 ± 20 ^d^	1470 ± 9 ^f^	−51.2 ± 1.0 ^b^	63.0 ± 1.1 ^b^

^a–f^ Different letters in the same column indicate a significant difference among the samples (*p* < 0.05).

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
