# Peer review of "Enhanced Interfacial Adhesion of Polylactide/Poly(ε-caprolactone)/Walnut Shell Flour Composites by Reactive Extrusion with Maleinized Linseed Oil"

_polymers, 2019, doi:10.3390/polym11050758_

Round 1

Reviewer 1 Report

The research article “Enhancing the Interfacial Adhesion of PolylactidePoly(ɛ-caprolactone)Walnut Shell Flour Composites by Reactive Extrusion with Maleinized Linseed Oil” by Sergi Montava-Jordà at all. investigates mechanical and thermomechanical properties of green composites consisting of a binary blend of polylactide (PLA) and poly(ɛ-caprolactone)composites modified with a waste filler in the form of ground walnut shell and maleinized linseed oil. The results are interesting and the analysis seems to be comprehensively performed. They also may prove useful for further research. However, a few issues need more attention, namely:

1.      Results (Figure 2 and 3) – Information about scale marks in the caption below the figure is necessary.

2.      Results (Page 8, Line 302) – Please change “anayslis” for “analysis”.

3.      Results (Page 11, Line 399) I suggest to include additional information about intensities of DTG which correspond to Tdeg and discussed in text. It may be useful for analysis of the degradation process.

4.      Results (Page 14, Line 490) – Please check sentence “…at 2,2 GPA at -80….”.

Author Response

1.      Results (Figure 2 and 3) – Information about scale marks in the caption below the figure is necessary.

The information about scale marks and also the magnifications at which the images were taken was already included in both figure captions.

2.      Results (Page 8, Line 302) – Please change “anayslis” for “analysis”.

This typo was corrected.

3.      Results (Page 11, Line 399) – I suggest to include additional information about intensities of DTG which correspond to Tdeg and discussed in text. It may be useful for analysis of the degradation process.

As the reviewer suggests, the effect of the fillers on the degradation rate of the biopolymers has been commented in the text. Please see page 13, lines 437-440.

4.      Results (Page 14, Line 490) – Please check sentence “…at 2,2 GPA at -80….”.

This sentence has been rewritten. Please see page 14, lines 502-505.

Reviewer 2 Report

This is a well written, clearly presented manuscript covering the important topic of producing more environmentally friendly polymer nanocomposites for the packaging industry. The manuscript deserves publication, but the authors should clarify/improve the following points:

-          Lines 54-55 – the sentence is not totally clear. Also, PLA is indeed a polymer with limited processability, given its poor thermal stability

-          Lines 65-68 - the two sentences seem somewhat contradictory. Please clarify

-          Line 127 – Why was this PLA/PCL ratio chosen?

-          Lines 132-134 - The screw speed used was surprisingly low. Why? With such a low screw speed it is difficult to accept that the (average?) residence time was approximately 1 min. What was the feed rate?

-          Figure 2 – Does the particle size histogram refer to individual particles, or are there also agglomerates? If yes, was there any particle dispersion upon extrusion and injection moulding?

-          Section 4 (Discussion) is just a summary of the discussions throughput the paper and could be easily removed.

-          Since the authors demonstrate in this work the double effect of plasticization and compatibilization of MLO, would it make sense to optimize its concentration? This could be a suggestion for further work

Author Response

-          Lines 54-55 – the sentence is not totally clear. Also, PLA is indeed a polymer with limited processability, given its poor thermal stability 

The sentence has been rewritten to avoid misunderstanding. Please see page 2, lines 54-55.

-          Lines 65-68 - the two sentences seem somewhat contradictory. Please clarify 

Following the recommendations of the reviewer, the sentence has been changed. Please see page 2, lines 64-66.

-          Line 127 – Why was this PLA/PCL ratio chosen?

 The selected ratio of PLA/PCL was chosen based on a previous study performed for this biopolymer blend at our lab. In particular, at ratios of 20-30 wt% PCL in PLA, these two biopolymers are immiscible but the improvement in ductile properties is quite good without compromising mechanical resistance properties. This information was added in the experimental part, section 2.2.

-          Lines 132-134 - The screw speed used was surprisingly low. Why? With such a low screw speed it is difficult to accept that the (average?) residence time was approximately 1 min. What was the feed rate?

The material was compounded in a co-rotating twin-screw extruder. The optimal screw speed varies as a function of the type and size of extruder, its L/D, and also screw configuration. In our facilities, the optimal processing conditions (in terms of material’s homogeneity and dispersion) for these materials are habitually achieved at speeds below 50 rpm. In relation to the residence time, this parameter is mainly determined by the feeding rate though the TSE screw rpms have some influence too. At a given feed-rate, residence time is gradually reduced with increasing screw speed. In our case, for an output of 5 kg/h and a processing speed of 20 rpm, it resulted in 1 min of residence time. We have already reported this in our previous studies, which are referenced in the present study. This information was also added in the experimental part, section 2.2.

-          Figure 2 – Does the particle size histogram refer to individual particles, or are there also agglomerates? If yes, was there any particle dispersion upon extrusion and injection moulding? 

The histogram refers to the size of individual particles, measured from several FESEM micrographs. Information about the dispersion and size of the WSF particles after melt processing can be seen in Figure 3 and it is discussed in Section 3.3.

-          Section 4 (Discussion) is just a summary of the discussions throughput the paper and could be easily removed.

In response to the reviewer suggestion, the discussion and conclusion sections have been shortened as much as possible and it has been refocused to main achievement of the study.

-          Since the authors demonstrate in this work the double effect of plasticization and compatibilization of MLO, would it make sense to optimize its concentration? This could be a suggestion for further work

As the reviewer suggests, the amount of MLO for this type of composition can be optimized in a near future work. However, our research group has already studied the optimal contents for MLO and other multi-functionalized vegetable oils in PLA-based materials, observing that the best performance is attained at 1-5 phr whereas higher concentrations habitually yields to saturation.

Reviewer 3 Report

The manuscript entitled “Enhancing the Interfacial Adhesion of Polylactide/Poly(ɛ-caprolactone)/Walnut Shell Flour Composites by Reactive Extrusion with Maleinized Linseed Oil". In this study, to continue and extend of our earlier research work dealing with MLO as a reactive additive to enhance compatibility in immiscible or low miscible green composites based on PLA. However, the manuscript as written is NOT suitable for publication in Polymers. However a REJECT and Resubmission of the paper is required before it can be accepted for publication in the Journal. The authors are advised to follow the comments given below very carefully in revising the manuscript. The manuscript has been marked for queries and they are listed below:

1.          Introduction, The literature review is poor and inadequate for the purposes of the paper. Generally speaking, the introduction section is a "teaching section" of the papers and it needs to provide the reader with the necessary background to be able to understand the problem, the aim, and the way adopted to achieve it. The introduction of this paper is too schematic and not useful in those senses.

2.          Conclusion. Poor comments.

3.          The English used in this manuscript is not well enough and hard to understand. Please kind check and revised.

4.          Figures are too blurry, and should be rearranged and reconstructed in order to make them more reader friendly.

5.          No statically results are shown in the table and discuss in the texts.

6.          In fact, the whole article seems to be meaningless since it didn't show enough discussion within the results and discussion section. Only the present of the results and few discussions were shown. Please kindly reorganized and rewritten the whole article, otherwise, this will be very ridiculous.

REFERENCES: the following indexing references should be cited: Optimization of Extrusion Variables and Maleic Anhydride Content on Biopolymer Blends Based on Poly (hydroxybutyrate-co-hydroxyvalerate)/Poly (vinyl acetate) with Tapioca Starch. CY Wu, WB Lui, J Peng Polymers 10 (8), 827

Author Response

1. Introduction, The literature review is poor and inadequate for the purposes of the paper. Generally speaking, the introduction section is a "teaching section" of the papers and it needs to provide the reader with the necessary background to be able to understand the problem, the aim, and the way adopted to achieve it. The introduction of this paper is too schematic and not useful in those senses.

In the opinion of the authors, the current Introduction section fully describes the necessary background for the study: first paragraph describes pollution originates from conventional plastics and the development of green composites as a plausible solution, second one introduces the potential of PLA and PLA/PCL blends, third one the opportunities of walnut shells to develop the green composites, fourth one reports about the use of MLO and reactive extrusion to improve the performance of the composites, and the last one summarizes the information that the study will provide to the reader. I do not understand what the reviewer means by the ambiguous term "teaching section".

2.          Conclusion. Poor comments.

The Discussion and Conclusion sections have been reduced and, again, they contain the necessary and most relevant information that can be obtained from the study.

3.          The English used in this manuscript is not well enough and hard to understand. Please kind check and revised.

Some typos were found and corrected. We have also rewritten some of the sentences to facilitate reading comprehension. Changes are now underlined in yellow and they can be tracked.  

4.          Figures are too blurry, and should be rearranged and reconstructed in order to make them more reader friendly.

The manuscript includes TIFF images at higher resolution (300 dpi). This issue should not be related to our manuscript.

5.          No statically results are shown in the table and discuss in the texts.

Statistical analysis of variance was performed for the data included in the tables. Please see Tables 2, 3, 4, and 5.

6.          In fact, the whole article seems to be meaningless since it didn't show enough discussion within the results and discussion section. Only the present of the results and few discussions were shown. Please kindly reorganized and rewritten the whole article, otherwise, this will be very ridiculous.

The results of the materials developed in this study are fully supported by a detailed set of characterization experiments and a complete explanation of the differnt phenomena and effects observed. This reviewer comment is, from our point of view, an opinion rather than a valid statement since he/she does not provide any technical information and it is even neither based on any scientific argument. Moreover, the words “meaningless” or “ridiculous” when referring to a manuscript seem totally inappropriate and unacceptable for a revision in a high-quality journal.

REFERENCES: the following indexing references should be cited: Optimization of Extrusion Variables and Maleic Anhydride Content on Biopolymer Blends Based on Poly (hydroxybutyrate-co-hydroxyvalerate)/Poly (vinyl acetate) with Tapioca Starch. CY Wu, WB Lui, J Peng Polymers 10 (8), 827

This reference has been added. Please see number 51.

Round 2

Reviewer 1 Report

Not all of the remarks listed in review have been applied, but I recommend this paper for publication.

Best regards 

Author Response

We performed all the requested changes and/or provided a response to each reviewer's comment.

Reviewer 2 Report

The authors replied satisfactorily to most of my comments

Author Response

(The authors gave the same response as above.)
